# The Effect of Expertise during Simulated Flight Emergencies on the Autonomic Response and Operative Performance in Military Pilots

**DOI:** 10.3390/ijerph19159141

**Published:** 2022-07-27

**Authors:** Sara Santos, Jose A. Parraca, Orlando Fernandes, Santos Villafaina, Vicente Javier Clemente-Suarez, Filipe Melo

**Affiliations:** 1Departamento de Desporto e Saúde, Escola de Saúde e Desenvolvimento Humano, Universidade de Évora, 7004-516 Évora, Portugal; sara.gsantos@hotmail.com (S.S.); orlandoj@uevora.pt (O.F.); svillafaina@unex.es (S.V.); 2Comprehensive Health Research Centre (CHRC), University of Évora, 7004-516 Évora, Portugal; fmelo@fmh.ulisboa.pt; 3Facultad de Ciencias del Deporte, Universidad de Extremadura, 10003 Caceres, Spain; 4Faculty of Sports Sciences, Universidad Europea de Madrid, 28670 Madrid, Spain; vctxente@yahoo.es; 5Grupo de Investigación en Cultura, Educación y Sociedad, Universidad de la Costa, Barranquilla 080002, Colombia; 6Laboratory of Motor Behavior, Faculty of Human Kinetics Technical, University of Lisbon, 1649-004 Lisboa, Portugal

**Keywords:** heart rate variability, military pilots, autonomic modulation, performance

## Abstract

Heart rate variability (HRV) and performance response during emergency flight maneuvers were analyzed. Two expert pilots (ages 35 and 33) and two rookie pilots (ages 25) from the Portuguese Air Force participated in this case–control report study. Participants had to complete the following emergency protocols in a flight simulator: (1) take-off engine failure, (2) flight engine failure close to the base, (3) flight engine failure far away from the base, and (4) alternator failure. The HRV was collected during all these maneuvers, as well as the performance data (the time it took to go through the emergency protocol and the subjective information from the flight simulator operator). Results regarding autonomic modulation showed a higher sympathetic response during the emergency maneuvers when compared to baseline. In some cases, there was also a higher sympathetic response during the emergency maneuvers when compared with the take-off protocol. Regarding performance data, the expert pilots accomplished the missions in less time than the rookie pilots. Autonomic modulation measured from HRV through portable devices can easily relay important information. This information is relevant since characterizing these maneuvers can provide helpful information to design training strategies to improve those psychophysiological responses.

## 1. Introduction

The military is one of the most demanding professions since warfighters must deal with real-life threats in various contexts [1]. In this line, aircraft pilots must face the fly combat requirement while supporting the extreme mechanical load of the aircraft, the hypoxia, and in this situation, being able to respond to any unforeseen event [2]. Pilots must endure a lot of stress during flight emergencies and still be able to give adequate responses to survive and avoid damaging the aircraft. For this reason, the capacity to work under these conditions is essential in this field, as is the ability to safely train and test the pilots [3,4,5,6,7].

The continuous and demanding requirements of different operations areas directly affect cognitive resources, being able to become a relevant handicap for this population [8]. Cognitive performance is crucial during these military operations. Impaired cognitive performance is considered responsible for most accidents during training and real battles [9,10,11]. In goal-oriented behavior, intrinsic, top-down attention allocation at the early stages of conflict monitoring allows a flexible response during the conflict detection process. It occurs through rapid prioritization and allocation of resources during the initial engagement and processing of external spatial stimuli. This will allow the pilot’s attention to engage with salient external warning cues while refocusing on task-relevant stimuli—a necessary feature of effective conflict detection [12,13,14,15,16,17,18,19,20,21,22]. 

The exposition of the demanding context of combats also produces an enormous emotional impact. In this line, fear is an emotional symptom of anxiety that occurs in response to a perceived threat and can trigger various defensive reactions. Fear can serve adaptive or maladaptive roles to either promote avoidance and escape from genuinely threatening situations or when attached to innocuous stimuli, can promote unnecessary stress. Generating appropriate defensive behaviors towards threats is essential to survival. Although many of these behaviors are “hard-wired”, they are also flexible. A hallmark feature of these circuits is their ability to activate the sympathetic branch of the autonomic nervous system, which relies on neural circuits and hormonal pathways such as adrenalin secretion to increase overall alertness leading us to action. Specifically, the hypothalamus-pituitary system and medial prefrontal cortex play pivotal roles in gating fear reactions (fight or flight reaction) and instrumental actions, mediated by the amygdala and nucleus accumbens, respectively [23,24,25,26,27]. 

The autonomic nervous system reactivity can be evaluated by the heart rate variability (HRV), which is a non-invasive method based on the analysis of successive heartbeats variation over an interval of time. The neural communication pathways interacting between the brain and the heart (through the endocrine system) are responsible for the generation of HRV [28,29,30,31,32]. When the HRV is reduced, the sympathetic activation predominates, indicating a reduced regulatory capacity to adapt to challenges such as exercise or stressors [33,34]. Thus, the HRV could be a biomarker of behavior flexibility or cognitive load. An optimal level of variability within an organism’s key regulatory systems is critical to the inherent flexibility and adaptability or resilience that epitomizes healthy function and well-being. While too much instability is synonymous with inefficient physiological functioning and energy utilization, too little variation indicates depletion or pathology. HRV is also an indicator of psychological resilience and behavioral flexibility, reflecting the individual’s capacity to adapt effectively to social or environmental demands. Furthermore, previous studies have shown an association between higher resting HRV and performance on cognitively demanding tasks requiring executive functions [4,34,35,36,37]. 

Previous studies in the military population and especially in pilots reported an increase in sympathetic activity during flights and combat maneuvers, as well as increased cortical arousal [4,5,7,38,39]. Low HRV typically reflected excessive sympathetic and/or inadequate parasympathetic modulation of heart rate, with reduced HRV observed during periods of emotional and physical stress (behavioral rigidity) and increased HRV during rest [4,6,7,37,40]. 

Different simulation training has been employed to improve the organic response and performance of pilots. Simulation training has roots in the aviation industry, with the first flight simulators built in the 1930s. The military, NASA, and commercial airlines further pioneered simulation techniques to improve pilot training and safety. The use of simulators in aviation has been proposed as the most economical way of training pilots, providing an excellent environment to cope with situations that are difficult to reproduce during real conditions and, therefore, an excellent transfer tool between training conditions and real aircraft. Aircraft pilots are routinely tested for proficiency and ability to handle emergencies. In addition, the use of flight simulators reduces the risk of catastrophic errors. Debriefing, which occurs at the end of the simulation, is a crucial aspect of learning, soliciting learner self-reflection and uncovering how they think about and approach problems. Interprofessional simulation is useful to understand and improve team dynamics and communication, and beyond training and education, simulation has real-world implications, including enhanced safety and quality. Previous studies investigated the psychophysiological demands of different basic civil aviation maneuvers in a flight simulator, showing that take-off and landing are some of the most stressful maneuvers. These results provide relevant information for training purposes [4,31,41,42,43,44]. However, there is a need for an article that analyses the psychophysiological response of pilots to emergency flight maneuvers.

Thus, the present research aimed to analyze the autonomic response and operative performance in different flight simulator emergency flight maneuvers. Expert and rookie pilots were included in this case–control report to investigate the influence of experience on the psychophysiological response of professional combat aircraft pilots. 

## 2. Materials and Methods

### 2.1. Participants

Four professional military pilots from the Portuguese Air Force participated in this control case report study (Table 1). The two pilots with more years of military service in the airbase (expert pilots -EP-) and the two pilots with fewer years of military service (tookie pilots -RP-) participated were chosen. The EP was assigned to duty in the airbase, one of their tasks being to teach the RP, while the RP were finishing up their masters in aeronautic military science in the airbase.

Measurements were made in March 2022 in Beja (Portugal). Expert pilots (33 to 35 years old) have more experience than novel rookie pilots (25 years old) with more than 10 years of differences in military service. Regarding monthly time spent in the flight simulator, rookie pilots had 180 min vs 60 spent by expert pilots. Regarding aerobic training, rookie pilots conducted more time practicing this training, whereas expert pilots conducted more strength training than rookie pilots.

### 2.2. Ethics Approval

The Évora University research ethics committee approved all the procedures. In addition, participants gave written informed consent, according to the Helsinki declaration (approval number: 21050), prior to participating in this study.

### 2.3. Procedures

The autonomic response of pilots was evaluated in 4 different maneuvers: Emergency 1 (E1)—take-off engine failure; Emergency 2 (E2)—flight engine failure close to the base allowing the pilot to land promptly; Emergency 3 (E3)—flight engine failure far away from the base not allowing the pilot to promptly land; Emergency 4 (E4)—alternator failure. In E1 the emergency protocol response starts during the take-off maneuver, on E2, E3, and E4 the pilots have time for the regular take-off maneuver, as well as flying before the emergency is activated by the simulator controller, leading to the emergency protocol response actions. 

For more details, in E1, E2, and E3 the engine of the plane stops working for any reason that might include one or any combination of the situations: low oil pressure indication, high oil temperature indication, or excessive engine vibrations. In E4 the network of components from the aircraft’s electrical system that generates electrical energy stops working during flight. In addition, in the E1 maneuver, the emergency protocol response starts during the take-off maneuver. If the engine failure occurs at takeoff, the pilot needs to choose if he can safely stop the takeoff or if the speed is already too much the only option is to choose an emergency turn route to land as soon as possible. In E2 and E3 the pilot needs to choose an emergency turn route to land as soon as possible since the aircraft is already flying. In E2 given the aircraft is closer to base and has just finished ascending they have less time to maneuver, in E3 since the aircraft is far away, they need to decide on the fastest route, or even if landing off base is an option. In E4 the pilot needs to check if components that transmit, distribute, utilize and store electrical energy are working and minimize the electrical energy spend as well as turn back to base to land. All participants underwent the same missions in the same order since the maneuvers could not be randomized. 

All the maneuvers were performed in a flight simulator (Epsilon - SEPS TB30). This flight simulator allows mimicking visual and instrument flight conditions. In this regard, a screen projected (in a 180° view) the simulated environment that coincides topographically with the real world. The cabin has real instruments which controlled all the movements of the simulated aircraft (see Figure 1). The mission lasted approximately 30 minutes and began at the same air base that the pilots serve.

### 2.4. Instruments

The time it took to complete every action was measured with a chronometer (stopwatch app from Huawei P20Pro), and the flight simulator controller appraised their performance relaying information on good protocol action choices or bad ones. Operative performance was measured during the simulation maneuvers.

Autonomic modulation was measured by the HRV in basal conditions (3 min) at a sitting position inside the simulator cabin prior to the simulation maneuvers as well as during the simulation maneuvers. Participants did not take any drug, alcohol, or other substance that could affect the nervous system 24 h before the protocol. Heart rate monitor Polar H10 [29,45] was used to measure HRV. Different HRV variables were extracted in the time domain, frequency domain, and non-linear measures. In the time domain, the mean heart rate (HR), the RR intervals, the standard deviation of all normal-to-normal RR intervals (SDNN), the percentage of intervals >50 ms different from the previous interval (pNN50), and the root mean square of successive differences RR interval differences (RMSSD) were analyzed [34,45]. In the frequency domain, the low frequency (LF) (ms2), the high frequency (HF) (ms2), and the ratio between LF/HF and the total power were calculated [34,46]. Lastly, non-linear measures, such as RR variability from heartbeat to short term Poincare graph (width) (SD1), RR variability from heartbeat to long term Poincare graph (length) (SD2), and Stress Index (SI—representing the degree of load on the autonomic nervous system) were also obtained [34,46,47]. Performance was measured by the time to complete the protocol of emergency response: after the flight simulator controller activated the emergency on the simulator, the pilots had to identify the emergency and choose the correct actions to respond. The time to complete the actions was measured with a chronometer.

## 3. Results

Table 2 shows the HRV and performance results of experts and rookie pilots. According to HRV results, the E1 (take-off with engine failure) seemed more demanding. This is determined by the higher values of heart rate and SD2 as well as lower values of RMSSD, SDNN, and total power compared to baseline. This could indicate an increase in sympathetic modulation as well as a reduction in parasympathetic modulation. In addition, these HRV variables indicated that E1 exhibited a higher autonomic modulation than the rest of the maneuvers (with the lowest values of RMSSD, SDNN, and total power as well as with the higher values of heart rate and SD2). However, a similar response, according to HRV variables, was found between rookie and expert pilots in the E1 maneuver. In the rest of the maneuvers and take-offs, experts showed higher sympathetic responses than rookie pilots. This can be induced by the results observed in the RMSSD and SDNN. 

Regarding performance, expert pilots showed higher results than rookie pilots except in the E2 maneuvers.

## 4. Discussion

This research aimed to analyze the effect of flight experience on the autonomic response and operative performance of professional combat aircraft pilots in a flight simulator emergency flight maneuver. Regarding the cardiovascular response of pilots analyzed, expert pilots presented lower heart rate values in baseline conditions. It could be related to aspects such as age (higher age is related to lower basal heart rate [48]) and the larger experience (that produces higher controllability and a lower feeling of uncertainty in this population, producing a lower sympathetic modulation and consequently lower heart rate [49]). Regarding the first take-off maneuver, the sympathetic response was higher than the other take-off maneuvers analyzed. This response could be related to the fact that the pilots did not know what they had to face in the first maneuvers, so the anticipation response they presented was higher [50]. The anticipatory anxiety response is a normal response, presenting an adaptive function since it prepares the organic systems (both physiologically and psychologically) to deal with a potential threat [1]. This organic response has also been evaluated in both military [51,52], and civil contexts where the level of contextual demand or stress is higher, such as commercial pilots [53,54], sports competitions [55,56], sanitary [49,50], or even academic events [57,58,59]. In this line, we should point out that the first incidence occurred only in the first take-off for all subjects, which is a source of additional stress to this anticipatory response [3], explaining this greater sympathetic modulation of the participants.

In the take-off of the three other emergency maneuvers, independently of their experience, all the pilots presented a higher parasympathetic modulation. The absence of emergency in these initial maneuvers and the experience of the first one must be the causes that explain this autonomic response. Analyzing these three emergencies, we found that the E2 (flight engine failure close to the base allowing the pilot to promptly land) presented the higher sympathetic response in both groups, while the RMSSD and SDNN presented the lower value of all the maneuvers analyzed. As we hypothesized, this emergency maneuver, which presented a higher possibility of returning to the air base, was associated with less time for the decision-making process and landing actions. However, it showed the largest stress response from all pilots. Comparing the results obtained in the emergencies analyzed in the present research with the previous literature, emergency pilots presented lower HRV and higher sympathetic modulation than in simulated combat flight maneuvers [7,60,61]. It seems that the existence of the constant uncertainty of an emergency, which was the simulated maneuvers evaluated in this article, produces a higher autonomic stress response in pilots than simulated air combat maneuvers, in which the pilot already foresees the final objective of these, which is flight combat. We can see how uncertainty is once again marked as a fundamental parameter that module the autonomic stress response, as has been postulated in previous works. Moreover, comparing the results obtained with real combat flights, we see that the cardiovascular response (heart rate) and sympathetic autonomic modulation are greater in real flight [3,62]. Lastly, although we do not verify this situation in the simulator, in the real flight situation, if there is a real perception of threat to physical integrity and a real risk of death, this consequently produces a greater stress response [40].

By analyzing the autonomic stress response of expert and rooky pilots, we found that experts presented a similar autonomic stress response (in some maneuvers, even higher sympathetic stress response than novel pilots). According to the habituation theory, this result would be expected since a rookie pilot with greater exposure to a stressful event should present habituation and develop a lower stress response [63]. This is supported by simulator time of exposure (180 min per month for the rookie pilots vs. 60 min per month for the expert pilots) which means that rookie pilots are more used to the simulator while expert pilots are more used to performing real flights. When we compare our results with other highly stressful events, we can see that when the stress level of the context is high, the appearance of a habituation response is more difficult [64,65]. In addition, in military contexts, it has been observed that soldiers with more experience and training have a greater psychophysiological response than those with less experience and training, which allows them to show greater performance in combat and enables them to achieve the mission objectives [66,67]. In this perspective, expert pilots showed higher performance than rookie pilots, although they presented a similar autonomic response. It is shown that the fact of having a higher capacity for information processing and action in these highly eliciting situations, where the psychophysiological stress response is very high, is what predicts the performance in these military contexts. In line with previous studies [68,69], it would be interesting to analyze how other conditions, such as the psychological profile of the subject, can affect this stress response and modulate the habituation response to stressors as high as those pilots and military are submitted to.

The main limitation of the present research was the low number of participants analyzed, which can be explained by the difficulty of accessing such a specific sample of elite pilots, as well as the protected involvement that surrounds these types of specialized environments. Future research with larger samples must help understand the psychophysiological stress response in these demanding contexts. In this approach, analyzing the influence of different behavioral factors such as psychological, physical activity, nutrition, and psychological profiles [70,71] may help to obtain a better comprehension of the stress response allowing for multidisciplinary interventions in this kind of population, in particular and in general populations exposed to large stress environments. Having more objective data on the pilot’s psychophysiological workload during flying tasks would be beneficial for the selection process concerning the different types of aircraft and to pilots, as well as for the missions assigned, and it could also be used as one more predictor of their performance [53,54].

## 5. Conclusions

The results of this research revealed that experimented and novel pilots presented increased sympathetic stress responses when submitted to different types of stressful situations such as the occurrence of aircraft emergencies during the flight. Nevertheless, these increases are similar and, in some cases, higher than those observed in experimented pilots during real flight situations. However, concerning flight competencies, experienced pilots showed higher performance in solving flight emergency tasks than novices.

## Figures and Tables

**Figure 1 ijerph-19-09141-f001:**
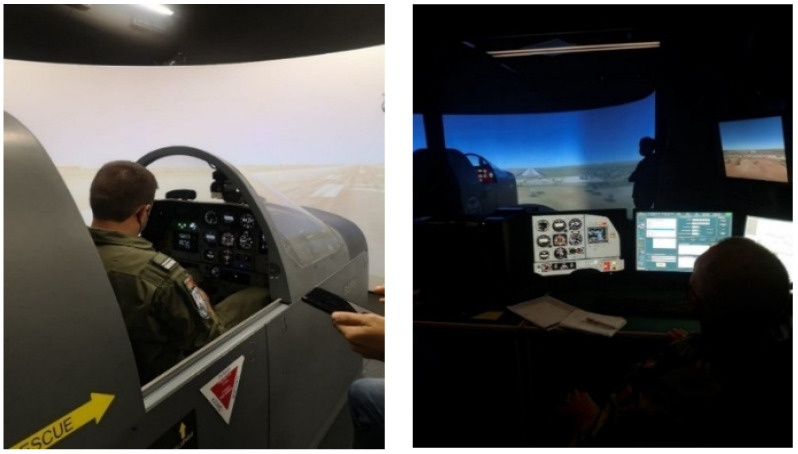
**Left**, the inside of the cockpit for the flight simulator Epsilon TB-30 and a pilot ready to start the protocol. **Right**, the inside of the control room and the setup device of the flight simulator operator.

**Table 1 ijerph-19-09141-t001:** Characteristics of military pilots.

Variable/Subjects Experience	EP1	EP2	RP1	RP2
Age (years)	35	33	25	25
Military service (years)	18	15	7	7
Height (cm)	183	175	185	175
Weight (kg)	88	80	80	62
Monthly periods in flight simulator (min)	60	60	180	180
Monthly periods in aerobic training (min)	720	1440	480	2100
Monthly periods in strength training (min)	720	240	150	120
Type of training	Crossfit	Long Distance Triathlete	Middle-distance running	Middle-distance running

EP1: Expert Pilot 1; EP2: Expert Pilot 2; RP1: Rookie Pilot 1; RP2: Rookie Pilot 2.

**Table 2 ijerph-19-09141-t002:** The results of the HRV in baseline and emergency maneuvers for the two EP and the two RP.

		Baseline	T1 and E1	T2	E2	T3	E3	T4	E4
MEAN HR (bpm)	EP	81.00 (±1.40)	94.50 (±3.50)	85.00 (±8.50)	88.50 (±3.50)	84.00 (±1.40)	90.00 (±8.50)	82.50 (±2.10)	86.50 (±3.50)
	RP	82.00 (±5.70)	102.00 (±9.90)	89.00 (±7.10)	93.00 (±9.90)	86.50 (±12.00)	89.50 (±9.20)	80.00 (±11.30)	85.00 (±7.10)
RMSSD (ms)	EP	21.50 (±6.60)	10.55 (±1.30)	19.40 (±4.80)	15.30 (±0.70)	14.65 (±0.60)	17.40 (±6.10)	21.90 (±2.40)	17.20 (±1.70)
	RP	38.70 (±19.10)	11.90 (±4.00)	27.40 (±17.40)	25.10 (±13.90)	30.15 (±21.60)	25.50 (±14.60)	37.45 (±26.80)	32.45 (±19.70)
LF (ms^2^)	EP	0.10 (±0.02)	0.072 (±0.02)	0.10 (±0.01)	0.06 (±0.02)	0.07 (±0.03)	0.08 (±0.00)	0.12 (±0.04)	0.07 (±0.04)
	RP	0.10 (±0.04)	0.10 (±0.07)	0.10 (±0.03)	0.11 (±0.02)	0.09 (±0.01)	0.09 (±0.01)	0.44 (±0.47)	0.11 (±0.02)
HF (ms^2^)	EP	0.21 (±0.08)	0.20 (±0.02)	0.19 (±0.05)	0.23 (±0.07)	0.18 (±0.04)	0.15 (±0.00)	0.16 (±0.01)	0.18 (±0.01)
	RP	0.21 (±0.06)	0.16 (±0.02)	0.18 (±0.02)	0.18 (±0.01)	0.16 (±0.01)	0.20 (±0.04)	0.16 (±0.010)	0.15 (±0.00)
POWER (ms^2^)	EP	1404.50 (±1368.30)	372.50 (±392.40)	659.00 (±408.70)	481.50 (±335.90)	545.50 (±368.40)	805.00 (±106.10)	1184.50 (±108.20)	725.50 (±150.60)
	RP	2666.50 (±2591.50)	393.00 (±239.00)	1263.50 (±1031.70)	1511.50 (±1219.80)	3231.00 (±3405.40)	1106.50 (±553.70)	3462.00 (±3900.40)	3639.50 (±3752.60)
SDNN (ms)	EP	33.15 (±17.20)	19.95 (±10.70)	29.35 (±1.60)	24.20 (±3.10)	23.05 (±9.30)	27.30 (±4.00)	32.60 (±1.10)	26.90 (±3.30)
	RP	46.45 (±16.10)	23.35 (±7.10)	39.80 (±23.60)	39.70 (±15.80)	54.65 (±37.80)	42.45 (±18.90)	58.45 (±39.20)	51.40 (±29.60)
SD2 (%)	EP	73.10 (±5.20)	76.15 (±8.30)	74.20 (±4.10)	74.90 (±1.70)	73.05 (±8.10)	74.50 (±3.70)	73.6 (±3.10)	74.45 (±4.60)
	RP	69.20 (±3.80)	78.95 (±0.90)	73.45 (±1.10)	75.65 (±3.50)	77.65 (±0.80)	76.55 (±2.90)	74.90 (±0.60)	75.10 (±1.00)
SI	EP	15.40 (±7.20)	23.10 (±0.90)	13.80 (±1.80)	15.10 (±0.90)	22.20 (±7.50)	17.90 (±3.60)	14.60 (±1.10)	13.90 (±2.90)
	RP	11.00 (±3.80)	21.90 (±0.60)	13.10 (±6.20)	13.50 (±4.00)	13.80 (±8.30)	13.50 (±5.60)	11.40 (±7.30)	10.50 (±4.80)
Performance (min)	EP	-	1.22 (±0.20)	3.68 (±0.70)	4.83 (±2.40)	0.32 (±0.00)	0.30 (±0.00)	2.88 (±0.50)	5.22 (±0.20)
	RP	-	1.60 (±0.80)	4.77 (±2.50)	4.31 (±1.10)	0.38 (±0.10)	0.92 (±0.60)	4.12 (±0.10)	10.88 (±0.50)

EP: expert pilots; RP: rookie pilots; T: take-off; E: emergency; RMSSD: the square root of the mean of the squares of the successive differences of the interval RR; LF/HF: low frequency (LF. 0.04–0.15 Hz) ratio (ms2)/high frequency (HF. 0.15–0.4 Hz) (ms2); total power: the sum of all the spectra; SDNN: standard deviation of normal-to-normal intervals; SD2: dispersion. standard deviation. of points along the axis of line-of-identity in the Poincare plot; SI: stress index; performance: time to solve the emergency (min).

## Data Availability

Not applicable.

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
