# Peer review of "The Effect of Expertise during Simulated Flight Emergencies on the Autonomic Response and Operative Performance in Military Pilots"

_ijerph, 2022, doi:10.3390/ijerph19159141_

Round 1

Reviewer 1 Report

Although the sample is very small, I believe that this is an original and well-conducted study. So, I recommend publication after these minor corrections:

1.                  Improve the title.

2.                  Materials and Methods

-          Line 112, Please, write in which period the study was conducted;

-          Line 113, Add a susection for committee approval.

3.                  Results

-          Please better explain the results in Table 2.

Author Response

Although the sample is very small, I believe that this is an original and well-conducted study. So, I recommend publication after these minor corrections:

Thank you very much for all your comments and proposals. We believe that the article has become much better, richer in content, and more noticeable to the reader.

  1. Improve the title.

Thank you very much for the comment we agree that the title should be changed and respecting your proposal we decided to change it to: “The effect of expertise during simulated flight emergencies on the autonomic response and operative performance in military pilots”. We believe that this new title is more in line with the work developed

  1. Materials and Methods

-          Line 112, Please, write in which period the study was conducted;

Thank you very much for the comment we added the phrase “Measurements were made in March 2022 at the military air base of Beja (Portugal).” (line 122)

-          Line 113, Add a subsection for committee approval.

Thank you very much for the comment, we created a subsection for committee approval: “2.2. Ethics approval” (2.2 line 132)

  1. Results

-          Please better explain the results in Table 2.

Thanks for your kind suggestion. In the results section, we have included more information, which we believe can better explain the content of table 2. (Lines 217 to 230)

Reviewer 2 Report

Dear Authors,

Thank you for your manuscript. The topic is interesting and important. The paper is well-designed and well-written and corresponds to journal formats. My initial concern about the topic was that it is too narrow and too specific for a public health journal, but then I saw that the paper was submitted to the special issue of case-clinic and case-control reports. So I have only a few minor comments about the paper.

The Introduction is well-written. The study aim is clear. A full-stop is missing at the end of the study aim.

More detailed characteristics of the pilots referencing Table 1 in the text would be beneficial.

For the readers, who are not familiar with the pilots' training, a more detailed explanation of emergency protocol and response actions would be beneficial (lines 124-125), as well as a more detailed presentation of the flight simulator (line 127). 

In the results, you present the data as means and standard deviations of the two rookie and expert pilots (Table 2). Maybe some non-parametric statistical tests could be applied to demonstrate significance in differences (e.g., Mann-Whitney U test to compare RP and EP, and Freedman test to compare the differences of the characteristics presented during different emergencies in the same group)?

Also, for me, the description of the results seems to be too scarce and incomplete. For example, such statements as "...the E1 (take-off with engine failure) seemed more demanding" and "...expert pilots showed higher results than the rookie pilots" need a more detailed explanation.

However, I congratulate you on your study!

Author Response

Thank you for your manuscript. The topic is interesting and important. The paper is well-designed and well-written and corresponds to journal formats. My initial concern about the topic was that it is too narrow and too specific for a public health journal, but then I saw that the paper was submitted to the special issue of case-clinic and case-control reports. So I have only a few minor comments about the paper.

The Introduction is well-written. The study aim is clear. A full-stop is missing at the end of the study aim.

Thank you for your constructive and valuable feedback. It has been included.

More detailed characteristics of the pilots referencing Table 1 in the text would be beneficial.

Thank you for your suggestion. We have included more information, in section 2. Materials and Methods / 2.1. Participants for better explain the content of table 1. (Lines 114 to 127)

For the readers, who are not familiar with the pilots' training, a more detailed explanation of emergency protocol and response actions would be beneficial (lines 124-125), as well as a more detailed presentation of the flight simulator (line 127).

Thank you for your suggestion. We put all the information requested in the procedure section (Lines 150 to 173). Often, as they are classified military areas with action secret protocols, we do not always have access to all the information, however, all the information that can be disclosed was given to us and is now in the article.

In the results, you present the data as means and standard deviations of the two rookie and expert pilots (Table 2). Maybe some non-parametric statistical tests could be applied to demonstrate significance in differences (e.g., Mann-Whitney U test to compare RP and EP, and Freedman test to compare the differences of the characteristics presented during different emergencies in the same group)?

Thank you very much for your proposal, which we consider totally relevant, however, as it is a Case-Control Report, and possibly because of the sample size, with the statistical analysis you propose, significant statistical values are not verified at any time.

Most CASE CONTROL studies present graphs and tables using immediate comparisons of results (for 1 subject) and means with standard deviation (short) for 2 subjects. As such, and respecting your proposal, we consider that it would not be pertinent to put the value of p, because really in a Case Report, it is not common to do and it could even be a mistake that will mislead the reader. It is our goal to continue working on this topic, and to increase the sample, as we consider the topic really interesting. And then yes, we will apply it according to your statistical proposals

Also, for me, the description of the results seems to be too scarce and incomplete. For example, such statements as "...the E1 (take-off with engine failure) seemed more demanding" and "...expert pilots showed higher results than the rookie pilots" need a more detailed explanation.

Thank you for your suggestion. We have included further information.

However, I congratulate you on your study!

Thank you very much for all your comments and proposals. We believe that the article has become much better, richer in content, and more noticeable to the reader.
